# Epigenetic Mechanisms Are Involved in the Oncogenic Properties of *ZNF518B* in Colorectal Cancer

**DOI:** 10.3390/cancers13061433

**Published:** 2021-03-21

**Authors:** Francisco Gimeno-Valiente, Ángela L. Riffo-Campos, Luis Torres, Noelia Tarazona, Valentina Gambardella, Andrés Cervantes, Gerardo López-Rodas, Luis Franco, Josefa Castillo

**Affiliations:** 1Institute of Health Research, INCLIVA, 46010 Valencia, Spain; fgimenovaliente@gmail.com (F.G.-V.); luis.torres@uv.es (L.T.); noetalla@incliva.es (N.T.); valen.gambardella@gmail.com (V.G.); andres.cervantes@uv.es (A.C.); gerardo.lopez@uv.es (G.L.-R.); pepa.castillo@uv.es (J.C.); 2Centro De Excelencia de Modelación y Computación Científica, Universidad de La Frontera, Temuco 01145, Chile; angela.riffo@ufrontera.cl; 3Department of Biochemistry and Molecular Biology, Universitat de València, 46010 Valencia, Spain; 4Department of Medical Oncology, University Hospital, Universitat de València, 46010 Valencia, Spain; 5Centro de Investigación Biomédica en Red en Cáncer (CIBERONC), 28029 Madrid, Spain

**Keywords:** alternative splicing, colorectal cancer, cell dissemination and metastasis, relapse, EZH2, G9A, histone methylation

## Abstract

**Simple Summary:**

The *ZNF518B* gene, which is up-regulated in colorectal cancer, plays a role in metastasis, but neither the mechanisms involved in this process nor the role of the different isoforms of the gene are known. Here we show that the ratio of these isoforms is related to the relapsing of the disease, and that the protein ZNF518B interacts with enzymes able to introduce epigenetic changes, which may affect the activity of many genes. We also report a list of genes affected in common by *ZNF518B* and by two of those related enzymes, namely, G9A and EZH2. An in-depth analysis of five of those genes revealed that ZNF518B is involved in the recruitment of the enzymes and in the deposition of the corresponding epigenetic marks. The results highlight the relevance of epigenetic changes in cancer development, and open the possibility of developing therapeutic approaches, as the introduction of epigenetic modifications is reversible.

**Abstract:**

The *ZNF518B* gene, which is up-regulated in colorectal cancer, plays a role in cell dissemination and metastasis. It encodes a zinc-finger protein, which interacts with histone methyltransferases G9A and EZH2. The expression of the two major mRNA isoforms 1 (coding for the full protein) and 2 was quantified by RT-qPCR in a cohort of 66 patients. The effects of silencing *ZNF518B* on the transcriptome of DLD1 and HCT116 cells were analysed by Clariom-S assays and validated by RT-qPCR. The recruitment of methyltransferases and the presence of H3K27me3 were studied by chromatin immunoprecipitation (ChIP). The ratio (isoform 2)/(isoform 1) negatively correlated with the relapsing of disease. The study of the transcriptome of DLD1 and HCT116 cells revealed that many genes affected by silencing *ZNF518B* are related to cancer. After crossing these results with the list of genes affected by silencing the histone methyltransferases (retrieved in silico), five genes were selected. ChIP analysis revealed that the recruitment of EZH2 is *ZNF518B*-dependent in *KAT2B*, *RGS4* and *EFNA5*; the level of H3K27me3 changes in accordance. G9A also binds *RGS4* and *PADI3* in a *ZNF518B*-dependent manner. The results highlight the importance of epigenetics in cancer and open a novel therapeutic possibility, as inhibition of histone methyltransferases may reverse the disease-linked histone marks.

## 1. Introduction

The human *ZNF518B* gene maps to chromosome 4 and, according to the data retrieved from the Ensembl genome browser (www.ensembl.org, accessed on 6 December 2020), possesses five exons and five splicing variants. The mRNA of the canonical isoform (ENST00000326756.4), further referred to as isoform 1, has 6954 nucleotides, and only part of the exon 5 is translated to yield a protein that is 1074 residues long in its unprocessed form. The ENST00000507515.1 mRNA (isoform 2) results from exons 2 and 3 skipping and a premature transcription termination of exon 4. It has only 527 nucleotides and may be putatively translated to give a product of 75 amino acids identical to those of the N-terminus of isoform 1. Both RNA-seq and RT-qPCR analyses have allowed us to check that the five isoforms are actually expressed in colorectal cancer (CRC) cell lines, although isoforms 1 and 2 account for most of the transcripts. Moreover, their expression level is higher in HCT116 and DLD1, which harbour *KRAS* mutations, than in the wild-type *KRAS*-containing Caco2, SW48 and RKO cell lines [1]. The canonical protein, translated from isoform 1 mRNA, contains three zinc-finger domains, located at 162–184, 190–213 and 1036–1058.

Genetic variations in the *ZNF518B* locus have been associated with gout [2,3,4] and with the accumulation of intramuscular fat [5], and epigenetic changes in the gene have been proposed as blood molecular markers of diabetes [6], but the exact function of the gene in these pathological processes is not known. A first molecular approach to the *ZNF518B* functions was reported by Lee and her colleagues, who found that the canonical ZNF518B protein activates the histone methyltranferase G9A encoded by *EHMT2/G9A* [7]. As the dysregulation of this enzyme has been associated with CRC [8], we further developed the initial work with CRC cell lines [1] and showed that *ZNF518B* is up-regulated in patients of colorectal cancer. By silencing the gene in CRC cell lines, it was demonstrated that its expression favours cell migration, invasiveness and other phenotypic variations, which suggests that the gene may be involved in the epithelial-to-mesenchymal transition (EMT) [9].

As mentioned above, the canonical isoform has three zinc-finger domains and, therefore, it is a likely DNA-binding protein. In this line, it has been proposed that ZNF518B may mediate the recruitment of G9A to the correct genomic loci [7]. Nevertheless, no information as to the possible DNA target sites for the *ZNF518B* product is available to date, and yet, it would be an interesting issue to explain the molecular basis of the effects of *ZNF518B* in CRC.

The method used in our studies on the up-regulation of *ZNF518B* in CRC patients [9] did not allow us to determine the level of the individual splicing variants. It is also worth examining this topic, as there are a plethora of data showing that aberrant splicing is involved in many types of cancer [10]. To mention some recent findings concerning CRC, it was shown that alterations in alternative splicing are involved in tumour cell proliferation [11,12,13,14,15,16] and invasiveness [15,17,18,19,20,21,22]. Consequently, the analysis of splice variants has been proposed as a tool for the diagnosis or prognosis of CRC [14,23,24,25,26] and the possibility of targeting specific variants or splicing modulators for therapeutic purposes has also been postulated [22,26,27,28,29].

In an effort to understand the molecular causes of the involvement of *ZNF518B* in CRC progression and EMT, in the present paper, we first examine whether the differential expression of isoforms 1 and 2 is related to any phenotypic characteristic of CRC patients, and then show our results, which found that the lower the ratio of isoform 2/isoform 1, the higher the probability of relapsing. Then, we describe a search for the genes whose expression is regulated by *ZNF518B.* In some of these genes, known as tumour suppressors, ZNF518B participates in the recruitment of G9A and/or EZH2 histone methyltransferases. The results show that epigenetic mechanisms are involved in the oncogenic properties of *ZNF518B.*

## 2. Results

### 2.1. Differential Expression of ZNF518B Isoforms in CRC Patients

We have previously reported that the major *ZNF518B* splicing isoforms 1 and 2 are expressed to a higher level in CRC tissues than in normal mucosa in a commercial cDNA array [9]. In the present study we present the results of the analysis carried out with a cohort of 66 patients (Appendix A). We found that isoform 2 mRNA is expressed to a significantly higher level than the canonical isoform 1 (Figure 1A).

Next, it was examined whether the expression of the individual isoforms 1 and 2 is related to any of the clinicopathological conditions of the patients. No significant relation was found, so we checked whether the ratio of isoform 2 to isoform 1 correlated with those clinicopathological characteristics. The results of this analysis are given in Figure 1B–H, which shows that the only significant correlation occurred between the isoform ratio and the relapse of patients (*n* = 10). The Kaplan–Meier curve of Figure 1I illustrates this correlation, showing that the disease-free survival of patients having a high isoform 2/isoform 1 ratio was significantly longer than that of patients with a low isoform ratio.

The classification of patients according to consensus molecular subtypes (CMS) could be done in 60 out of the 66 patients of the cohort, and the ratio isoform 2/isoform 1 was significantly low in the CMS1 patients (Figure 1J).

The DNA sequencing results obtained by a custom-targeted next-generation sequencing panel [30] indicated that activating *KRAS* mutations were present in 32 out of the 60 patients analysed, followed by mutations in *TP53* and *APC* (Appendix A). Interestingly, although the expression of *ZNF518B* depends on the presence of *KRAS* mutations in some CRC cell lines [1], the gene expression and the isoform ratio are independent of the mutational state in patients. This issue is discussed below.

The results presented here, especially those concerning the absence of a correlation between the isoform ratio and the invasiveness of tumour (Figure 1C,D,G,H), suggest that the differential expression of *ZNF518B* isoforms is not responsible for the effects of the gene on the dissemination of CRC cells [9], although the isoform ratio correlates with relapsing. Therefore, we next explored whether the interactions between the expression of *ZNF518B* and that of other genes may account for the previously reported phenotypic effects. As these effects were investigated by silencing the gene [9]; we first complemented those studies by analysing the consequences of the overexpression of *ZNF518B*, as described in the next section.

### 2.2. Overexpression of ZNF518B in RKO Cells

The *ZNF518B* gene is expressed in some CRC cell lines, while in others transcripts of the gene are either absent or present to negligible amounts. With respect to the cell lines, HCT116, DLD1 and some derivatives of the latter rank among the first group of lines, while RKO, SW48 and Caco2 fall into the second category [1]. The phenotypic consequences of knocking down the gene with specific small interfering RNAs (siRNAs) in HCT116 and DLD1 cells have been described [9]. In this section, we report the consequences of overexpressing *ZNF518B* in RKO cells. The transfection mentioned in Material and Methods efficiently increased the level of *ZNF518B* mRNA (Figure 2A), and the immunocytochemical experiment of Figure 2B, apart from checking the overexpression of the gene at the protein level, showed that ZNF518B localizes to the nucleus.

The overexpression of *ZNF518B* resulted in an increased migration capacity of RKO cells, as revealed by the transwell assays (Figure 2C). Invasiveness also increased after overexpressing the gene (Figure 2D) and the capacity of adhering to type I collagen was also enhanced after *ZNF518B* overexpression (Figure 2E). These results were exactly the opposite of those obtained after silencing the gene [9], and therefore, both sets of results were in agreement. Although the knocking down of the gene did not affect the colony forming capacity [9], overexpressing the gene resulted in an increased capacity of colony formation (Figure 2F). It is possible that the extremely large increase in *ZNF518B* expression (Figure 2A) could be responsible for this apparent contradiction.

### 2.3. Functional Interactions of ZNF518B

As mentioned above, the physical interaction of the ZNF518B protein with the methyltransferase G9A has been described, and the same authors found that other histone methyltransferase, namely EZH2, is also a binding partner of ZNF518B [7], but the possible genetic interactions involving *ZNF518B* have not yet been examined. The presence of ZNF518B in nuclei (Figure 2B), is compatible with its proposed role as a DNA-binding protein. Taking into account that many zinc-finger proteins are transcriptional factors, it can be wondered whether ZNF518B plays that role. Should this occur, changes in the expression of *ZNF518B* would affect the expression of other genes. To explore this possibility, we analysed the global transcriptome of the CRC DLD1 and HCT116 cell lines to check how it is affected by silencing *ZNF518B*.

The analysis was carried out by using a Clariom-S assay, as described under Materials and Methods, with three independent controls and three *ZNF518B*-silenced cell cultures with both cell lines. When only the genes changing their expression by a factor ≥ 2 were considered, knocking down the expression of *ZNF518B* caused the up-regulation of 163 and 276 genes in DLD1 and HCT116, respectively, as well as the down-regulation of 282 genes in DLD1 and 322 genes in HCT116 (Appendix A). The global analysis of the data, including the results of the three replicates from each cell line, with both unmodified and silenced *ZNF518B*, is shown in Appendix A. The principal component analysis clearly revealed the differences in the transcriptome between DLD1 and HCT116 cells, and how the triplicate samples, after the knocking down of the *ZNF518B* cluster between them, clearly separated from the control samples in the graph (Appendix A). The heat map of Appendix A summarises the results of the transcriptomic data. The automated clustering based on the similarities of genetic profiles was done by separating the cell lines according to the transcriptomic modification generated by silencing *ZNF518B.* In this way, it facilitated the comparison between the transcriptome profile of both cell lines in the presence and absence of the knocking down of *ZNF518B*. Table 1 shows the gene ontology of the altered genes that are related to cancer-linked biological processes, such as proliferation, EMT, focal adhesion, cell cycle, etc., which are related to the phenotypic changes associated with both overexpression and silencing [9] of the gene in CRC cell lines. These genes (*n* = 374), represent a large majority of those affected by silencing *ZNF518B*.

We next validated the results of the above analysis by RT-qPCR. To do this, some genes were selected. The selection was based on two criteria, namely, the greater silencing-induced fold change in expression in both cell lines, and the involvement in CRC progression, according to the literature data. The results (Figure 3) agree, from a qualitative point of view, with those of the transcriptomic analysis (Appendix A).

### 2.4. Molecular Mechanisms Involved in the Oncogenic Properties of ZNF518B

The ZNF518B protein, a putative DNA-binding factor, might affect CRC development by activating genes with oncogenic properties. The genes *MOB1A*, *RBL2*, *S100A14*, *SRPX*, *GINS1*, *BABAM1*, *CERK*, *TOMM34* and *DDAH1*, whose changes in expression as a consequence of knocking down *ZNF518B* were validated by RT-qPCR (Figure 3), were candidates to check whether ZNF518B bound to their promoter. Actually, these genes promote the growing of cells, and it is possible that ZNF518B acts as a transcriptional factor, activating their expression. This question was addressed by chromatin immunoprecipitation (ChIP) analysis. Chromatin from DLD1 and RKO cells was immunoprecipitated with the anti-ZNF518B antibody. The latter cells, in which *ZNF518B* is not expressed [1], were used as a negative control. The resulting DNA was amplified by PCR with the primers located around 400–500 bp upstream of the first exon (see Materials and Methods). Amplification was not detected in any case, and we think that the antibody used, albeit suitable for immunocytochemistry (Figure 2B), was not appropriate for ChIP.

A second mechanism to account for the oncogenic behaviour of *ZNF518B* is that it represses tumour suppressor genes. This function might be carried out by recruiting the inhibitory histone methyltransferases EZH2 and/or G9A, which physically interact with ZNF518B. In fact, an enhanced level of *EHMT2* expression results in decreasing the disease-free survival of CRC patients, as revealed by the analysis of The Cancer Genome Atlas (TCGA) database (Appendix A). To check whether that second mechanism was actually operative, we first searched for the genes affected simultaneously by silencing *ZNF518B* and the genes coding for those histone methyltransferases.

### 2.5. Genes Affected in Common by the Silencing of ZNF518B, EHMT2 and EZH2 in HCT116 Cells

A list of genes whose expression is altered by silencing *EHMT2* or *EZH2* in HCT116 cells (Appendix A), reported in common by STAR-edgeR and Bowtie2-edgeR pipelines, was recovered from GSE108210 RNA-seq raw data as described under Materials and Methods. These results were contrasted with the experimental list from the Clariom-S analysis of the genes altered by *ZNF518B* silencing in HCT116 cells by the procedures described under Materials and Methods. In the Clariom-S array, 575 genes were reported as differentially expressed (*p*-value < 0.01) by *ZNF518B* knocking down (Appendix A). In the RNA-seq results, 792 genes were reported as affected by *EHMT2* knocking down and 1228 genes by *EZH2* knocking down, as shown in Appendix A (*p*-value < 0.01). These results were contrasted to find the genes that were differentially expressed in common after silencing *ZNF518B*, *EHMT2* and *EZH2* (Appendix A and Figure 4). Out of the 575 genes affected by *ZNF518B* knocking down, 28 were also affected by *EHMT2* silencing, 61 were also influenced by *EZH2* silencing and the expression of 29 genes was simultaneously influenced by the three genes under study (Appendix A).

There were 118 genes affected in common by *ZNF518B*, *EZH2* and/or *EHMT2* (Appendix A). Among these genes, we first selected those inhibited by *EZH2* and/or *EHMT2*. Then, we crossed the resulting list with that of the genes inhibited by *ZNF518B*. Obviously, the final list contained all the candidate genes to be inhibited by *ZNF518B* in combination with *EZH2* and/or *EHMT2*. Finally, by analysing the published effects of these genes, we selected those displaying suppressor properties. In this way, the genes selected for ChIP analysis were: *PADI3*, a tumour suppressor in CRC [31], *ZDHHC2*, a suppressor of metastasis in hepatocellular carcinoma [32] and *RGS4*, which inhibits the growth of human breast carcinoma cells [33]. Two other genes, namely *EFNA5* and *KAT2B*, were also selected for ChIP analysis in view of their potential oncological relevance, although the cut-off value used to construct the list of Appendix A did not allow for their inclusion. The first one of these two additional genes has been previously reported as a suppressor in CRC [34,35], while the second one codes for the histone acetyltransferase PCAF, which has been described as an inhibitor of hepatocellular carcinoma [36]. The inclusion of *KAT2B* was further substantiated by the fact that it is one of the more connected nodes in the protein–protein interaction network (Appendix A) obtained by in silico analysis, as described under Materials and Methods.

The consequences of silencing *ZNF518B* on the expression of these genes were first validated by RT-qPCR in HCT116 cells (Figure 5A), in which the in silico analysis of the effects of silencing *EZH2* and *EHMT2* was done. As a further verification, we compared the expression levels of these genes in RKO cells transfected with pCMV3-*ZNF518B* or with the empty plasmid. As these genes were activated by silencing *ZNF518B*, it was expected that the overexpression of the latter resulted in the inhibition of the selected genes, and the experimental results confirmed this idea (Figure 5B).

### 2.6. Recruitment of Inhibitory Histone Methyltransferases by ZNF518B

The rationale to check the second possible mechanism to explain the oncogenicity of *ZNF518B* (see above) was that, if any of the selected genes is inhibited by the recruitment of EZH2 or G9A by ZNF518B, the silencing of the gene would result in a decrease of the binding of the methyltransferase over the gene and, consequently, would result in a diminished presence of the repressive mark on the neighbouring nucleosomes (Figure 6A).

For ChIP analyses, chromatin was immunoprecipitated either with anti-EZH2 or anti-G9A, and the resulting DNA was amplified with primers, which define amplicons of 100–150 bp (see Materials and Methods). These amplicons were located either at the promoter or the control regions, as annotated in the Ensembl database (Appendix A). Many of these regions actually bind zinc-finger factors and, therefore, it is reasonable to think that these sequences may also be targets for ZNF518B. The results obtained with the antibody against EZH2 are given in Figure 7A. The *PADI3* gene was not included in the study because the in silico analysis revealed that its expression was not altered by *EZH2* silencing. Therefore, the results shown in Figure 7A refer to 11 genomic regions, namely, the promoters of *KAT2B*, *EFNA5* and *ZDHHC2*, the first exon of *EFNA5* and seven control sequences: three from *RGS4*, two from *EFNA5* and one from *KAT2B* and *ZDHHC2.*

Barski et al. [37] showed that the repressive mark H3K27me3 extends all over the silenced genes, although it peaks in the neighbourhood of the transcription start site. To check whether this circumstance also occurred in our experiments, we included in our analysis a region from the 3’ flank of one of the genes, namely *EFNA5.* In 7 out of the 11 regions analysed, 4 of *EFNA5*, 2 of *KAT2B* and 1 of *RGS4*, the silencing of *ZNF518B* resulted in a significant reduction, or even in a non-detectable binding of EZH2. This suggests that the histone methyltransferase was recruited to these regions by ZNF518B. In other two cases, namely, the two of the *RGS4* control regions, the ChIP assay showed that EZH2 actually bound the corresponding sequences, but the binding was not influenced by ZNF518B. Obviously, other DNA-binding factors must be involved in the recruitment of the histone methyltransferase. Finally, in the remaining two cases, the promoter and the control region of *ZDHHC2*, the knocking down of *ZNF518B* resulted in a significant increase of EZH2 binding. We do not know the reasons for this behaviour, which is discussed later on.

Next, we examined whether the changes in the binding of EZH2 correlated with the level of H3K27me3 in those genomic regions, because this trimethylation results from the action of EZH2 [38]. The results of the ChIP analysis with anti-H3K27me3 are also shown in Figure 7A. In three genomic regions the presence of H3K27me3 could not be analysed, because the low signal obtained in the ChIP analysis precluded obtaining reliable information. In the remaining cases, the silencing of *ZNF518B* resulted in the disappearance of the epigenetic mark. Of note, this also occurred in the promoter of *ZDHHC2*, in which the presence of the enzyme EZH2 varied in the opposite sense. Nevertheless, it has to be kept in mind that EZH1, also an interaction partner of ZNF518B [7], shares the same catalytic specificity with EZH2.

Finally, the influence of the *ZNF518B* knocking down upon the binding of G9A to the selected genes was examined. The experiment was limited to *RGS4* (three control regions), and to the promoter of *PADI3*. The expression of the three remaining genes was not affected by *EHMT2*, as revealed by the in silico study, and therefore, the ChIP analysis was not carried out on these genes. The results are given in Figure 7B, which shows that the signal obtained for G9A was much less intense than that of EZH2. Apart from the possible differences in antibody effectiveness, it has to be mentioned that the binding of G9A to ZNF518B may be transient, and probably only a small proportion of complexes exist at any given time [7]. The binding of G9A to the three examined regions of *RGS4* and to the promoter of *PADI3* was ZNF518B-dependent. The results from the ChIP analyses allowed us to check the hypothesis that the oncogenicity of *ZNF518B* involves the recruitment of repressive histone methyltransferases to some suppressor genes (Figure 6B). This issue is further discussed below.

## 3. Discussion

The present research was undertaken with the aim of exploring the molecular mechanisms involved in the oncogenicity of *ZNF518B*, which has been shown to favour tumour cell dissemination in CRC. To address that issue, the role of alternative splicing isoforms of the gene and the consequences of the genetic and/or physical interactions of the gene and its product were investigated.

Many cancer-related genes experience alternative splicing, but most of the studies aimed at finding novel genes as disease biomarkers centre on the expression of their canonical isoforms. Instead, we focused on the different isoforms and on the ratio of the two major ones. The individual expression of these major isoforms of the gene, although increased in tumours when compared to normal adjacent tissues [9], did not correlate with the clinicopathological characteristics of the patients. Nevertheless, when the ratio isoform 2/isoform 1 was considered, a significant increase in disease-free survival occurred in patients with higher ratios (Figure 1I) and relapsing preferentially occurred in patients with lower ratios (Figure 1E). It can be wondered why their ratio was related to relapsing, while the levels of individual isoforms were not. The existence of mRNA corresponding to isoform 2 is beyond all doubt [1,9], but the actual existence of the putative product of its translation (a polypeptide 75 amino acids long) remains elusive. The influence of the isoform ratio on disease-free survival may then be ascribed to competitive effects at the mRNA level. In the event that the isoform 2 protein actually existed, a competition with the canonical isoform in the activation of G9A (see below) could not be discarded, as fragments from the N-terminus of ZNF518B, which lack the zinc fingers of the protein, are still able to activate the methyltransferase [7]. Albeit the actual causes that link relapsing with the isoform ratio are not fully understood, but this ratio might constitute a biomarker for CRC recurrence. This idea supports the approach used to construct Figure 1, in which we show the relationship between the isoform ratio and the clinicopathological characteristics of the patients. It is also interesting to note that the *ZNF518B* isoform ratio is related to the CMS classification; as shown in Figure 1J, the subtype 1 is characterised by a lower survival after recurrence [39]. In a previous in silico analysis of RNA-seq databases, and in the subsequent experimental validation, we found that the selection of the alternatively spliced isoforms of *ZNF518B* and the whole gene expression depended on the presence of the G13D *KRAS* mutation [1]. The present results, however, show that the expression of the whole *ZNF518B* and of its isoforms, as determined in a large cohort of patients, takes place irrespective of the mutational state of *KRAS*; these results are, therefore, more reliable.

The *ZNF518B* overexpression experiments described here (Figure 2) are in close agreement with those obtained by silencing the gene [9]. They were carried out in RKO cells, because in these cells the basal expression of the gene is negligible (Figure 2A) and, therefore, the effects of overexpression were more apparent. In the genes selected to clarify the molecular mechanisms of the *ZNF518B* oncogenic properties, the levels of their mRNA changed in an opposite way after overexpressing or silencing *ZNF518B* (Figure 5).

From the Clariom-S analysis described here, it may be concluded that the physical and/or genetic interactions of *ZNF518B* extend to a large number of genes. To restrict ourselves to the genes included in Figure 3, it can be mentioned that *SLC2A4RG*, which is repressed by *ZNF518B*, has recently been validated as tumour suppressor of human glioma [40], while *PTBP2*, also inhibited by *ZNF518B*, favours metastasis in CRC by activating the transcription of *RUNX2* [41]. Among the genes activated by *ZNF518B*, *CERK* is associated with highly aggressive human breast cancer [42] and *GINS1* expression correlates with proliferation of breast cancer cells [43] and with poor prognosis in hepatocellular carcinoma [44]. The mention of these genes, among those affected by *ZNF518B* (Appendix A), serves as an example that might explain the phenotypic consequences of the overexpression or silencing of *ZNF518B*. Nevertheless, it can be advanced that the role of *ZNF518B* in CRC is complex. At any rate, the basis for the oncogenicity of *ZNF518B* may involve either the activation of some protooncogene or the inhibition of a suppressor gene. Taking into account that ZNF518B possesses three zinc-finger domains, it is reasonable to think that it may act as a transcriptional activator binding to some protooncogene. ChIP experiments to detect this possibility failed because, as mentioned above, the available antibodies were not suitable for this type of analysis.

It was reported by Maier et al. [7] that the ZNF518B protein physically interacts with the histone methyltransferases EZH2 and G9A. Both enzymes act as “writers” of H3K27me3 and H3K9me2, respectively, which are considered as repressive marks [37]. Therefore, we reasoned that the oncogenicity of *ZNF518B* may be explained in part by its canonical protein binds, through its zinc-finger domains or control sequences of some tumour suppressor genes tethering any of the methyltransferases. To check this hypothesis, we carried out ChIP analyses using antibodies against the methyltransferases. The results obtained showed that the recruitment of EZH2 was *ZNF518B*-dependent in three genes, namely, *KAT2B*, *RGS4* and *EFNA5*, and that the changes in the binding of the methyltransferase were followed by the expected changes in the level of H3K27me3 (Figure 7A). In the case of *EFNA5*, our analysis covered the promoter, including a CCCTC-binding factor (CTCF) element, the first exon and the 3’ flank. The level of H3K27me3 peaked around the transcription start site and was maintained in both flanks of the gene, when the gene was repressed, i.e., in the absence of *ZNF518B* silencing. These results are in accordance with the findings of Barski et al. [37].

G9A also binds *RGS4* and *PADI3* in a *ZNF518B*-dependent manner (Figure 7B). Therefore, the above-mentioned hypothesis seems to be valid in most of the regions of the four mentioned genes. The implication of *EFNA5* in CRC was first reported by Wang et al. [34], and recently, it was shown that it is involved in CRC cell dissemination and metastasis; thus, it has been proposed as a potential therapeutic target in CRC [35]. Therefore, the functional relationships between *ZNF518B* and *EFNA5* are particularly interesting, as the latter gene may mediate the acquisition of the metastatic phenotype triggered by the former. Figure 6B summarises how the results obtained fit a mechanism in which ZNF518B tethers histone methyltransferases EZH2 and G9A to some tumour suppressor genes and the enzymes introduce repressive marks. This allows for triggering dissemination and the metastasis of tumour cells.

The results obtained with *ZDHHC2* seem to be contradictory, because knocking down *ZNF518B* resulted in an increase of EZH2 recruitment, and yet, the H3K27me3 mark disappeared. This mark may be introduced by EZH1, a member of the PRC2 complex, which also interacts with ZNF518B [7].

Although the value of the results presented here was limited by the size of the cohort, they highlight the importance of epigenetic factors in cancer development. This issue has been often addressed in recent years (for an early review, see [45]), and the reversibility of epimutations allows for using epigenetic drugs in cancer treatment. In the present instance, the expression of *ZNF518B* was regulated by epigenetic factors [9], and our results showed that its mode of action involved epigenetic modifications of histones. Further, several clinical trials using inhibitors of EZH2 are currently active [46], although caveats have been raised against the use of inhibitors of chromatin-modifying enzymes [47,48]. Obviously further investigation is needed, but the present work widens the net in the search for finding novel therapeutic approaches to control CRC progression by using epigenetic tools.

## 4. Materials and Methods

### 4.1. Human CRC Samples

We prospectively recruited patients diagnosed with localised colon cancer between October 2015 and October 2017 at the Hospital Clínico Universitario de Valencia, Spain. Follow-up computer tomography scans were done every 6 months for 5 years. Tumour tissues from 66 patients (see Appendix A for their clinicopathological characteristics) were collected before any treatment, immediately after surgery. Fresh samples were analysed and sequenced by next-generation sequencing for somatic mutations in the 29 genes that are recurrently mutated in colorectal cancer, as previously described [30], and the remaining tissues were stored at −80 °C until used. Formalin-fixed, paraffin-embedded tumour tissues from the surgical specimens were deposited at the INCLIVA biobank. CMS classification [39] was performed using a new 33-gene CMS assay and a rank-based classifier based on NanoString technology. As previously described [39], 100 ng of total RNA from paraffin-embedded tissues was used to perform the assay, using a 33-gene set derived from the original CMS study custom nCounter platform-based biomarker assay [49] and a centroid- and gen-based rank-based classifier method. Here, centroids refer to a summary statistic defining expression of each gene in each subtype. To predict disease-free survival, the “nearest CMS” classification was selected.

All the procedures were carried out in accordance with the Declaration of Helsinki and subsequent relevant guidelines from the World Medical Association and the European Union. The study was approved by the Ethical Committee of the Hospital Clínico Universitario de Valencia (No. 2018/237).

### 4.2. Cell Culture

The human CRC cell lines HCT116 (ATCC CCL-247) and DLD1 (ATCC CCL-221) were grown in McCoy’s 5A medium (Sigma, St. Louis, MO, USA). The cell line RKO (Horizon Discovery, Cambridge, UK) was grown in Dulbecco’s modified Eagle’s medium (DMEM). In every case, the media were supplemented with 10% foetal bovine serum (GE Healthcare), 1% L-glutamine (Biowest) and 1% penicillin/streptomycin (Biowest). To guarantee the continued quality of the cell lines used in this study, a short tandem repeat DNA profiling of the cells was performed. Genomic DNA was extracted at different steps of the study and the identity of the cells were confirmed by Bioidentity (Elche, Spain), complying with the international standards of authentication (ANSI/ATCC). Genomic DNA was routinely checked at the cell culture service of the Transcription Unit of Central Unit of Medical Research (UCIM; Central Research Unit, INCLIVA—Faculty of Medicine) to discard mycoplasma contamination by using the Mycoplasma Gel Detection kit (#90021 Biotools B&M Labs, Madrid, Spain) according to the manufacturer’s instructions.

### 4.3. ZNF518B Overexpression and Knocking Down

Overexpression was carried out by the transient transfection of RKO cells, which do not express the gene to a detectable level under normal conditions [1]. The plasmid used was pCMV3-*ZNF518B* (SinoBiological #HG27388-UT, Wayne, PA, USA) and the corresponding empty plasmid was used as the control. Plasmids were amplified in *Escherichia. coli* and purified with QIAprep Spin Miniprep Kit (Qiagen, #27104, Hilden, Germany) according to the manufacturer’s protocol. The purified plasmids were quantified with a Nanodrop 2000 spectrophotometer (Thermo Fisher Scientific) and stored at −20 °C until used. One day before transfection, 3 × 10^5^ RKO cells were plated in six-well plates and let grow in a medium without antibiotics until 80% confluence was reached. Then, a mixture of 4 µg of plasmid and 10 µl of Lipofectamine 2000 Reagent (Invitrogen, # 11668-027, Carlsbad, CA, USA) was added in the Opti-MEM Medium (Thermo Fisher # 31985047, Waltham, MA, USA). After 6 h, the medium was changed for maintenance up to 24 h with a standard medium.

Knocking down of the gene was achieved with siRNA, as described elsewhere [9].

### 4.4. Phenotypic Analyses of Transformed Cells

Cell migration and invasion assays, as well as the colony formation and adhesion to type I collagen-coated plates assays, were carried out exactly as described elsewhere [9], with cells transfected with both pCMV3-*ZNF518B* and the empty plasmid.

### 4.5. Immunocytochemistry

Immunocytochemical detection of ZNF518B in RKO cells, transfected either with pCMV3-*ZNF518B* or with the empty plasmid, was carried out after fixing the cells with 4% paraformaldehyde and permeabilising them with 2% Triton X100. Additionally, 2.5% normal horse serum (Vector, # S2012, Burlingame, CA, USA) was used as blocking solution for 1 h. Then, cells were incubated with 1/200 of a diluted ZNF518B antibody (Sigma, # HPA031216). An anti-rabbit secondary antibody (Dako Agilent, # P0448, Santa Clara, CA, USA) was used at 1/100 dilution and the slides were treated with 3,3’-diaminobenzidine (DAB) chromogen solution (Dako Agilent, # 3467, Santa Clara, CA, USA). The reaction was stopped for the subsequent haematoxylin–eosin staining.

### 4.6. Microarray Analysis of Gene Expression

RNA was obtained as described elsewhere [1] after 72 h of siRNA transfection and it was further processed at the Transcription Unit of Central Unit of Medical Research (UCIM) of the University of Valencia. After retrotranscription, cDNA was hybridised to an Affimetrix Clariom-S human array. Three independent biological replicates were carried out in the two cell lines HCT116 and DLD1 to ensure the reliability and reproducibility of the results. The analysis of the data was carried out by the Transcriptome Analysis Console (TAC) bioinformatics program (Thermo Fisher Scientific). To obtain a list of genes with significant changes (*p*-value < 0.05) between the control and silencing conditions, a filtering algorithm was applied and the fold-change of the expression between cells with knocked-down and normal *ZNF518B* was calculated through the log_2_ of the replicates average. Functional annotation was performed by using the software provided by the manufacturer.

### 4.7. In Silico Analysis of the Effects of EHMT2 and EZH2 Silencing

The GSE108210 RNA-seq data, corresponding to HCT116 cells with silenced *EHMT2*, (HCT116 si_G9A), silenced *EZH2* gene (HCT116 si_EZH2) and the control HCT116 (HCT116 si_Non-target) were downloaded from the SRA database [50]. The quality of the data was verified by using fastqc Version 0.11.8 [51]. Trimming was carried out with FASTX-Toolkit v0.014 (http://hannonlab.cshl.edu/fastx_toolkit/, accessed on 6 December 2020). The good-quality reads were aligned with the reference human genome (GRCh38) by using both STAR v2.7 [52] and Bowtie2 [53]. The differential expression analysis (si*EHMT2* or si*EZH2* vs. siControl) was done with the bioconductor package edgeR [54]. The genes reported in common by the STAR-edgeR and Bowtie2-edgeR pipelines were considered as differentially expressed. Finally, the results of this analysis were compared with those experimentally obtained by the Clariom S assay (see above) in HCT116 cells with silenced *ZNF518B*. The code for the in silico analysis is included in the Appendix A.

### 4.8. Protein–Protein Interaction Network

The protein interaction information was obtained from the STRING database v11.0 [55]. This database provides a confidence score (from 0 to 1), obtained from the estimated likelihood of each of the annotated interactions to be biologically meaningful, specific and reproducible [55]. Only connections obtained from experimental studies, publicly available databases and text mining with a minimum confidence score of 0.5 were included. The protein interaction network was analysed by Cytoscape v. 3.7.1 [56] and the unconnected nodes were excluded from the network.

### 4.9. Chromatin Immunoprecipitation (ChIP)

Chromatin isolation was carried out as previously described [57]. For the ChIP analysis, chromatin was sonicated to an average fragment size of 450–500 bp. Immunoprecipitation was done, as required with anti-ZNF518B (Sigma, HPA031216), anti-G9A (Abcam #ab40542) or anti-EZH2 (Abcam #ab191250), following the previously reported procedures [57,58]. At least two independent ChIP experiments were carried out for each antibody. Quantitative PCR was done, as mentioned below.

### 4.10. Quantitative PCR Analysis

RNA isolation and retrotranscription, as well as the RT-qPCR determination of the expression of the whole *ZNF518B* gene and of its individual isoforms 1 and 2 with specific primers, were carried out as previously described [1]. The sequences of these primers, as well as those used for the RT-qPCR validation of the data obtained from microarrays, or retrieved from databases by in silico analysis, are given in Appendix A. All the PCRs were done in triplicate and quantification was carried out by the 2^−ΔCt^ procedure, using the *ACTB2* gene, coding for β-actin as the standard. The primers used in the ChIP analyses are given in Appendix A. In this case, the 2^−ΔCt^ procedure was also followed with ΔCt = Ct(input)-Ct(immunoprecipitated). From this value, the ΔCt obtained without the antibody was subtracted. At least three technical replicates were carried out in each PCR.

### 4.11. Statistical Analyses

The statistical analyses were carried out according to the number of variables to be compared. The distribution of the data was first analysed by the normality tests of Shapiro–Wilk and Kolmogorov–Smirnov. The equality of the variances was analysed by the Levene test. To construct the Kaplan–Meier curves for disease-free survival using the data from the TCGA database, the procedure described elsewhere [59,60] was followed. When our experimentally determined isoform ratio was used, data were categorised into high and low groups according to the median values. Kaplan–Meier curves and pairwise comparisons were performed with R package ‘survminer’.15 and GraphPad Prism v.8.2.1. When the comparison of data involved only two variables, the Student’s t test was used. For comparisons involving more than two variables, Kruskal–Wallisor ANOVA tests were used depending on the data normality. Univariable and multivariable analyses were performed using Cox regression with R package ‘survival’.16. All statistical analyses were carried out with R V.3.6.1.17. All *p*-values below 0.05 were considered significant.

## 5. Conclusions

In agreement with the phenotypic changes observed after the silencing and overexpression of *ZNF518B*, we conclude that the expression level of the putative transcriptional factor alters the expression of genes involved in CRC progression and metastasis. Moreover, changes in the ratio of *ZNF518B* splicing isoforms are involved in the recurrence of the disease. Epigenetic mechanisms are involved in the oncogenic properties of the gene. Our results show that the ZNF518B protein recruits the histone methyltransferases EZH2 and G9A on several tumour suppressor genes and the resulting repressive marks on histone H3 contribute to reducing the expression level of these genes.

## Figures and Tables

**Figure 1 cancers-13-01433-f001:**
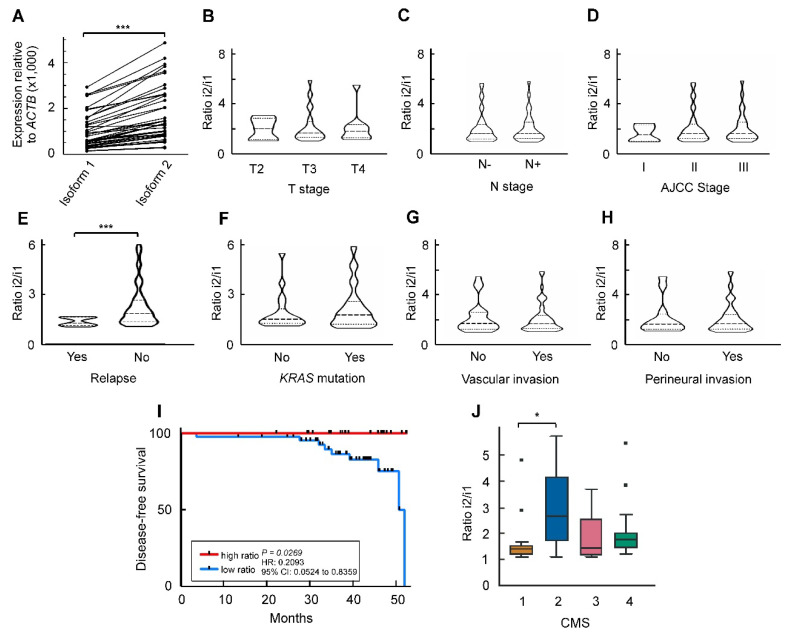
Expression of mRNA corresponding to the two main isoforms of *ZNF518B* to show that the isoform ratio correlates with the relapse of patients. (**A**) Differences in isoform level in tumour samples (a cohort of 66 patients was analysed). (**B**–**H**) Violin plots showing the changes in the ratio between isoform 2 and isoform 1 mRNA with respect to the clinicopathological properties of the patients’ cohort. In panels B and C, T and N stages are indicated in the conventional way. (**I**) Disease-free survival of patients as a function of the isoform 2/isoform 1 ratio. Patients were categorised in high- or low-ratio groups according to the median value of the ratio in the cohort. (**J**) Correlation of the ratio between isoform 2 and isoform 1 with the consensus molecular subtypes (CMS) classification. *, *p* < 0.05; ***, *p* < 0.001.

**Figure 2 cancers-13-01433-f002:**
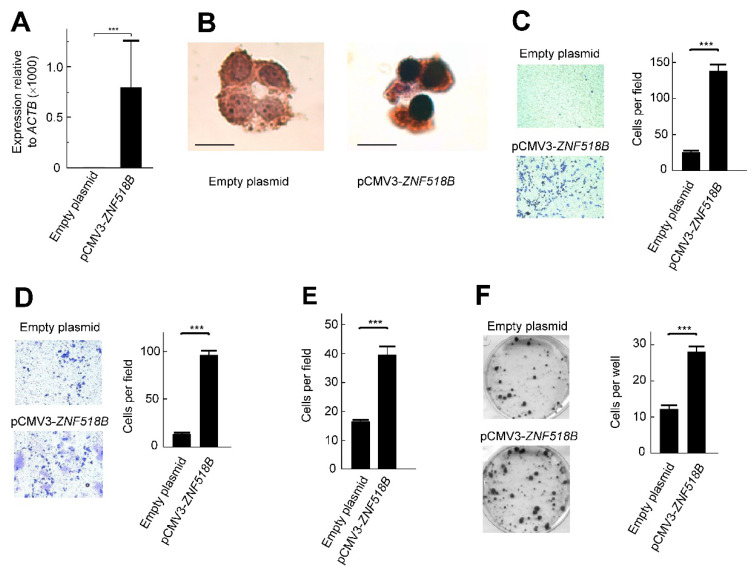
Overexpression of *ZNF518B* in RKO cells shows the nuclear location of ZNF518B and an increase in cell migration, invasiveness and adhesion to type I collagen. The cells were transfected with pCMV3-*ZNF518B*, or with the empty plasmid. (**A**) RT-qPCR analysis of the overexpression of *ZNF518B* in RKO cells. (**B**) Immunocytochemical analysis; size bar, 20 μm. (**C**) Transwell migration assay. A photograph of representative plates is shown (**left**) and the counting values (six different areas) were averaged (**right**). (**D**) Effects of overexpression of *ZNF518B* on the invasiveness of RKO cells. The assay was carried out as in (C), but through a Matrigel layer. (**E**) Effects of overexpression of *ZNF518B* on the adhesion of RKO cells to type I collagen-coated plates. The total number of cells was counted as in (C). (**F**) Colony formation assays carried out in six-well plates as described under Materials and Methods. Two independent experiments (each one with three independent transfections) were done with RKO cells. The photograph of representative plates and the averaged quantification of the number of colonies in each of the three wells are given. Statistical analyses were carried out by Student’s *t*-test. ***, *p* < 0.001.

**Figure 3 cancers-13-01433-f003:**
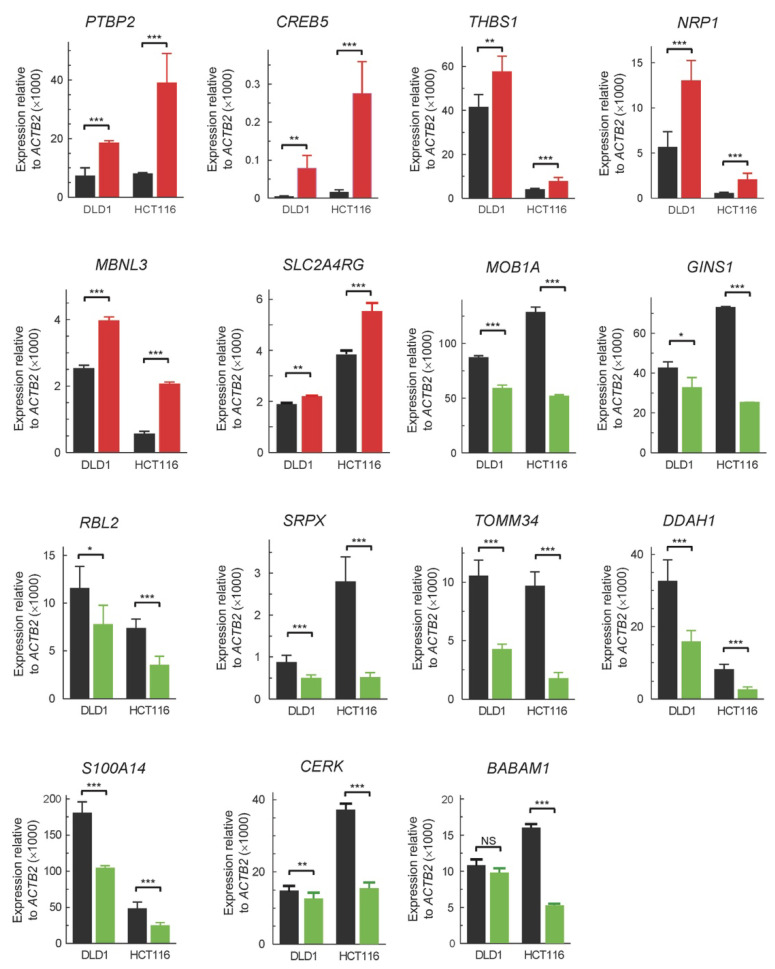
RT-qPCR validation of the changes in expression in some genes selected among those detected as affected by silencing *ZNF518B* in the Clariom-S analysis (see Appendix A). The analysis was done in both DLD1 and HCT116 cells. The black bars correspond to control cells, transfected with scrambled siRNA, while the coloured bars depict the results obtained with cells transfected with si-*ZNF518B.* *, *p* < 0.05; **, *p* < 0.01; ***, *p* < 0.001; NS, non-significant.

**Figure 4 cancers-13-01433-f004:**
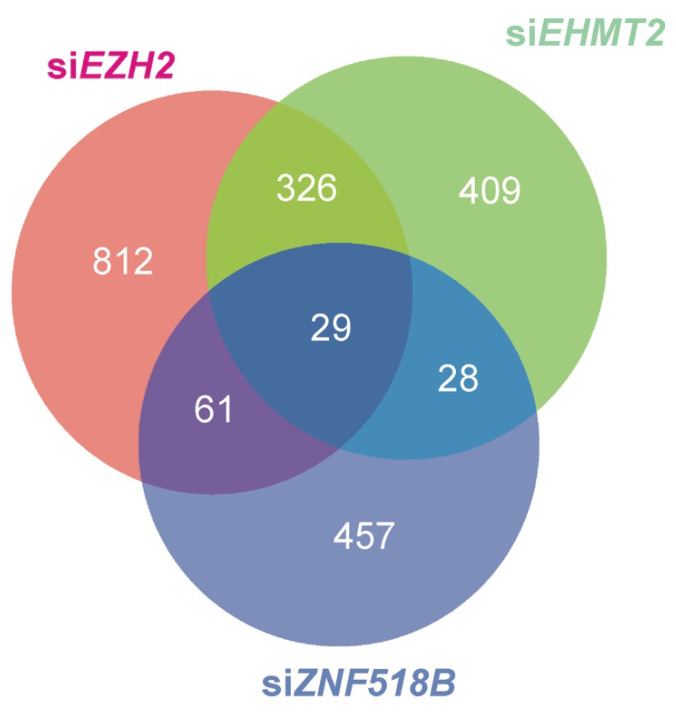
Venn diagram showing the genes whose expression is affected in common by the knocking down of *ZNF518B*, *EZH2* and/or *EHMT2* in the colorectal cancer (CRC) HCT116 cell line.

**Figure 5 cancers-13-01433-f005:**
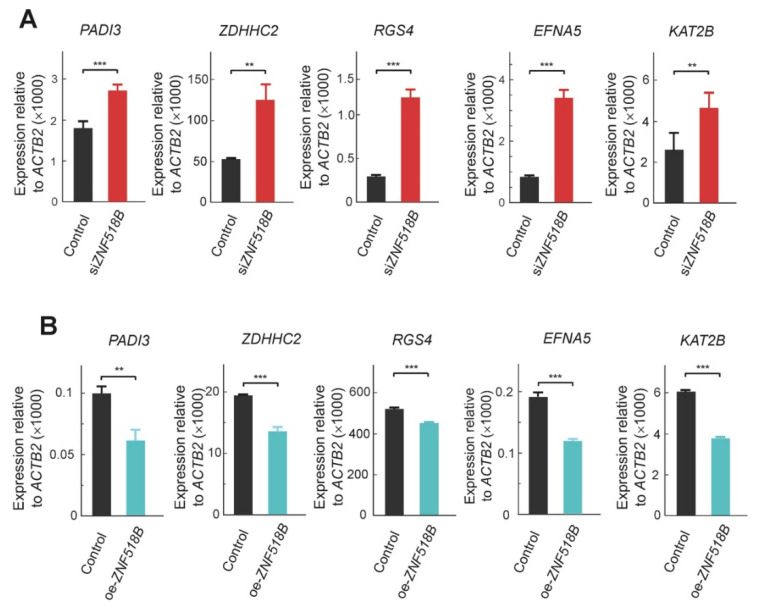
RT-qPCR validation of the changes in expression of the genes analysed for the binding of histone methyltransferases. The figure shows that *ZNF518B* actually inhibits the expression of these genes. (**A**) Changes in expression in HCT116 cells caused by the knocking down of *ZNF518B*. (**B**) Changes in expression in RKO cells caused by overexpressing *ZNF518B*. **, *p* < 0.01; ***, *p* < 0.001; NS, non-significant.

**Figure 6 cancers-13-01433-f006:**
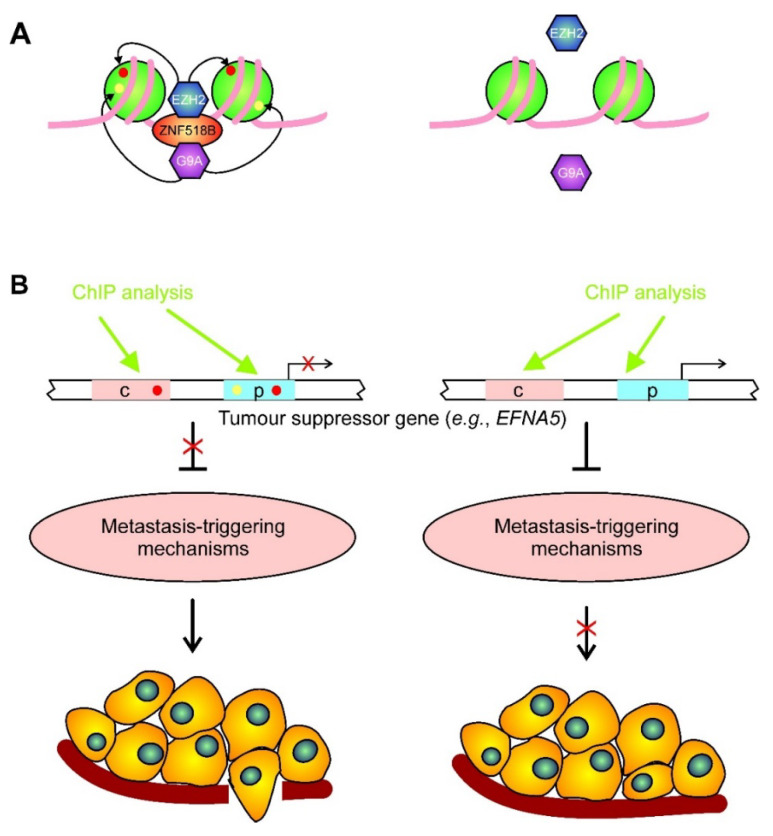
A possible mechanism for *ZNF518B* oncogenic role. (**A**) Schematic drawing of the starting hypothesis for the chromatin immunoprecipitation (ChIP) experiments. ZNF518B binds a specific DNA target in a tumour suppressor gene and recruits a histone methyltransferase (EZH2 or G9A), which catalyses the introduction of the repressive marks (H3K27me3, red ovals, or H3K9me3, yellow ovals, respectively) in the histones of the neighbouring nucleosomes (left). In the absence of ZNF518B, the histone methyltransferases are no longer recruited, and the repressive marks are not introduced (right). (**B**) The presence or absence of the methyltransferases and/or of the repressive epigenetic marks, either in the promoter (p) or in some control region (c), determines the expression of a suppressor gene. These epigenetic features can be experimentally detected by ChIP analysis. The expression of the suppressor gene, in turn, controls the triggering of mechanisms that eventually result in the dissemination and metastasis of tumour cells as a consequence of *ZNF518B* activity.

**Figure 7 cancers-13-01433-f007:**
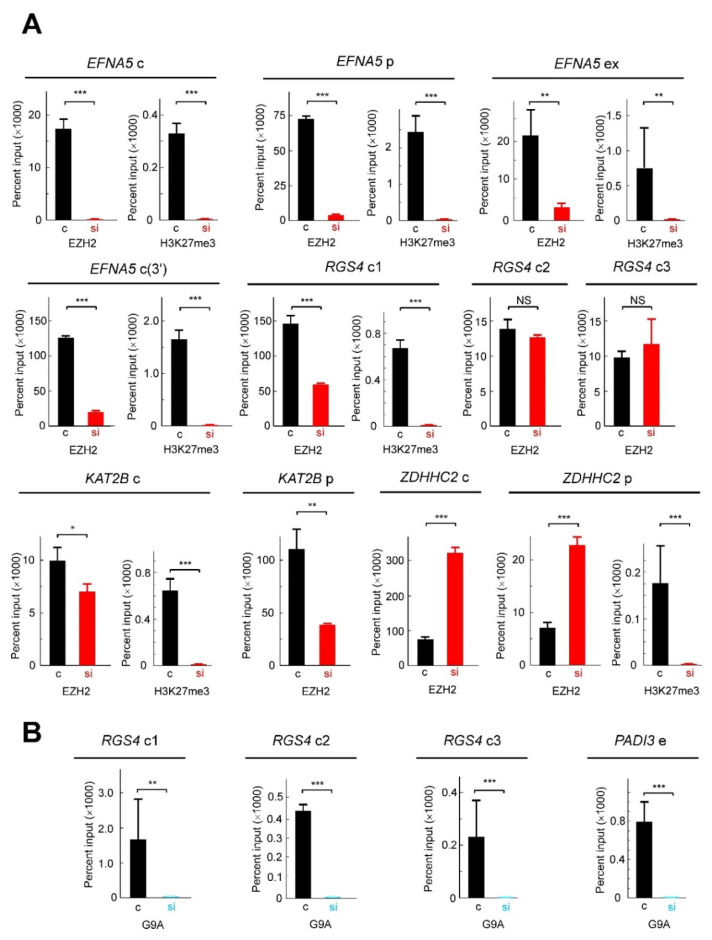
ChIP analysis of the presence of histone methyltransferases and of histone methylation in selected regions of genes affected by *ZNF518B* silencing. (**A**) Presence of EZH2 and the resulting mark (H3K27me3) in several regions of the selected genes. In some instances, the signal obtained in the immunoprecipitation with anti-H3K27me3 was too low to yield accurate results and the corresponding panel was omitted. (**B**) Presence of G9A in several regions of the selected genes. ChIP experiments were carried out with HCT116 cells in which *ZNF518B* was knocked down by transfecting with specific siRNA (si) or in control cells, transfected with scrambled siRNA (c), as indicated below the bar charts. The genes and their corresponding regions analysed (c, control regions; *p*, promoter; e, enhancer; ex, exon) are indicated over the charts. See Appendix A for the location of these regions. *, *p* < 0.05; **, *p* < 0.01; ***, *p* < 0.001.

**Table 1 cancers-13-01433-t001:** Gene ontology of the alterations induced in the transcriptome of DLD1 and HCT116 cells by silencing the *ZNF518B* gene.

Biological Pathway	Up-Regulated Genes (*n*)	Down-Regulated Genes (*n*)
EGFR pathway	14	14
Signalling VEGFR2	16	27
PI3K-AKT pathway	26	17
MAPK pathway	21	18
WNT pathway	8	9
TGF-β signalling pathway	13	13
RAS pathway	16	8
Biological processes	Up-regulated genes (*n*)	Down-regulated genes (*n*)
Cell cycle	9	20
EMT in CRC	12	9
Apoptosis	8	12
Histone modification	4	13
Focal adhesion	20	9
Focal adhesion via PI3K-AKT-mTOR	25	13
Only the genes related to cancer-linked processes are included.		

## Data Availability

Data is contained within the article or Appendix A.

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
