# Peer review of "Epigenetic Mechanisms Are Involved in the Oncogenic Properties of ZNF518B in Colorectal Cancer"

_cancers, 2021, doi:10.3390/cancers13061433_

Round 1
Reviewer 1 Report
Although many concerns of original review have been taken care of in the revised version of manuscript, it still needs minor edits (eg. replace 'paper' with 'study' in line 95; delete 'found' in line 102 and many more) and revision. Figure legends include a lot details of materials and methods but a very limited take home message for each panel. For a better presentation, it would be better to include a take home message for each panel of figure instead of mentioning the name of technique/assay only.
Author Response
Thank you very much for your suggestions. Accordingly, we have made the following changes in the manuscript:
1) The edits you mentioned and some others of our own have been introduced.
2) Some figure legends have been modified to shorten them, and to include a take home message. The latter has been introduced in the captions to Figs. 1, 2, 5 and 7 (the latter corresponding to the results of ChIP analyses, which were previously depicted as Fig. 6B). Please, note that a novel Fig. 6 is now incorporated, as suggested by Reviewer #2. This figure, as well as Figs. 3 and 4 do not include take home messages, as we think their captions are themselves explanatory.
Reviewer 2 Report
Specific comments to the authors
The authors Francisco Gimeno-Valiente et al. of the submitted manuscript „Epigenetic mechanisms are involved in the oncogenic properties of ZNF518B in colorectal cancer” studied the oncogenic potency of ZNF518B on colorectal cancer (CRC) in the relation to epigenetic regulative mechanism. Based on their intensive bio-integrative investigations the authors could demonstrate that (i) the recruitment of EZH2 is dependent of ZNF518B in KAT2B, RGS4 and EFNA5 and that (ii) the binding of G9A to RGS4 and PADI3 shows a ZNF518B-dependent manner, too. The authors postulated that the ZNF518B-epigenetic-axis in CRC show novel therapeutic possibilities via reversing the disease-linked histone marks.
Overall, the manuscript gives very interesting aspects of oncogenic mechanism of ZNF518B in CRC via the epigenetic players EZH2 and G9a. The manuscript (including presentation) is mostly comprehensible and convincing. The methods are mostly described. Although the results and discussion are clear presented, some minor changes must be performed by the authors (see specific comments) to improve the manuscript.
In conclusion, the presented data are very interesting. Nevertheless, the mentioned specific comments (see below) should be incorporated in the manuscript before accepting.
Specific comments
Abstract: The final conclusion should definitively include how the findings could be used for new therapeutic strategy.
Introduction: Please give more information about the relationship between KRAS mutations and the expression of the ZNF518B genes.
Results:
# Figure 1: What was the rationale for using the isoform ratio in the figure 1I? The KRAS mutations is very high which could be related to patient selection bias (?).
# Figure 2: Please add a magnification bar for figure 2B.
# Functional interactions of ZNF518B: please specify the sentence “To address this issue, we analysed the global transcriptome of the CRC DLD1 and HCT116 cell lines to check how is affected by silencing ZNF518B.”.
# Figure 3 to 5: An additional in-silico-analysis via STRING (Search Tool for the Retrieval of Interacting Genes/Proteins) should be performed for the investigated genes to show and to predict protein–protein interactions.
Discussion: Regarding to the complete study, the definitive role of the isoform analysis in the manuscript is not clear. As mentioned for the abstract, the final conclusion is largely speculative and should be more specified by the authors. Due to the complexity of the findings a summarized “mechanistic” figure could support the reading and understanding. The low case load for the expressional analysis of ZNF518B in CRC should be adequately mentioned for the limitations of the manuscript.
Author Response
First of all, we wish to thank you very for your review, which have aided to improve our manuscript. Following your suggestions, we have introduced the modifications mentioned below.
1) A short explanatory sentence of the therapeutic possibilities of our results has been included at the end of the abstract.
2) A more detailed account of our previous results concerning the relationships between KRAS mutations and expression of ZNF518B is now given in the Introduction.
3) Although the second paragraph of section 2.1 already outlined the rationale for using the isoform ratio in Fig. 1, you are right in that this question might be better explained. Consequently, we have introduced a novel sentence in this sense at the beginning of the Discussion (2nd paragraph).
4) We have obtained the patients’ data from a prospective cohort, as specified in section 4.1 and, therefore, there are no possibility of bias relative to KRAS mutational state of patients. Moreover, KRAS mutations occur in roughly 40% of CRC patients and in our cohort the percentage is around 50%. Taking into account the limited number of patients, that figure does not significantly deviate from the accepted data.
5) Size bars have been included in Fig. 2B.
6) The redaction of the first paragraph of section 2.3. Functional interactions of ZNF518B, has been modified. We think it is now clear why we analysed the effects of silencing ZNF518B on the transcriptome of those cell lines.
7) The STRING analysis you mentioned has been done and incorporated as Supplementary Figure S4. We are very indebted to your suggestion, because this analysis has provided us with a novel argument to justify the inclusion of KAT2B in the ChIP experiments. A short sentence in this sense is included in section 2.5.
8) In its present form, the manuscript includes a figure (Fig. 6), which summarizes the ChIP results and their possible mechanistic consequences. To draw this figure, we have used a modified version of the previous figure 6A, and have added a novel panel proposing a plausible mechanism for the role of ZNF518B. The results of the ChIP analyses (former Fig 6, B and C) are now given in the Figure 7.
This manuscript is a resubmission of an earlier submission. The following is a list of the peer review reports and author responses from that submission.
Round 1
Reviewer 1 Report
Title: Epigenetic mechanisms are involved in the oncogenic properties of
ZNF518B in colorectal cancer
In this paper Francisco Gimeno-Caliente and coworkers show that the zinc-finger protein codified by the ZNF518B gene and up-regulated in colorectal cancer (CRC) interacts with histone methyltransferases G9A and EZH2, thus introducing epigenetic changes which affect the activity of many malignancy-related genes. In a cohort of 66 CRC patients the authors have shown the the isoform ratio of the two different forms of ZNF518B correlates with the disease-free survival of the patients. Through a series of different approaches, ranging from RT-qPCR to ChIP analysis and various in silico validations, the authors have shown that several tumor suppressor genes (PADI3, a tumor suppressor in CRC; ZDHHC2, suppressor of metastasis in hepatocellular carcinoma; RGS4, growth inhibitor in breast carcinoma; EFNA5, tumor suppressor in CRC; KAT2B, inhibitor of histone de-acetylases in hepatocellular carcinoma) are upregulated upon siRNA silencing of ZNF518B, evidences that directly suggest an activity of ZNF518B in maintaining a low expression of onco-suppressor molecules. The results highlight the relevance of epigenetic changes in cancer development and open the possibility to develop new therapeutic approaches in the reversible control of ZNF518B-dependent epigenetic modifications.
In the opinion of this reviewer this is a very important study, whose results are likely to go beyond the tumor system used by authors and extend to other tumors overexpressing histone methyltransferases. Overall the paper is well written, the experimental design is clear and conclusions are logical. The cited literature is comprehensive. I rise only the following minor concerns:
- In the legend of figure 2, line 140: “The assay carried out as in C”: change to “the assay was carried out as in C”
- At the beginning of the Discussion, line 303: “ZNF81B”: change to “ZNF518B”
- Materials and Methods, line 368: “its mode of action involve”: change to “its mode of action involves”
Reviewer 2 Report
The manuscript by Gimeno-Valiente et al. describes some molecular aspects that involve the ZNF518B gene in clinical and phenotypic aspects of CRC cells.
The manuscript based on a previously published paper in the Scientific Report journal where the authors characterized the gene, its isoforms, and its involvement in cell migration and invasiveness in CRC.
Although potentially interesting the major manuscript' finding is a slight but significant correlation between ZNF518B isoforms2/1 ratio with tumor patients samples that relapsed. But this result is not clearly stated since no evident information in the main body of how many patients relapsed (in a cohort of stage I, II, and III CRC should not be a large number), a Kaplan-Meier curve should be more appropriate resuming the important characteristics of the groups (patients' follow-up). Then the authors do not satisfactorily explain why the expression does not correlate with KRAS mutations in light of their previous publications. The epigenetic part of the study is merely described and represented by difficult to follow tables (2 and 3), since controls results are not clearly outlined. The gene ontology analysis is too speculative and does not well support the rationale "...some genes were selected.." to validate the analysis by RT-qPCR.
In summary, the manuscript presents a few molecular mechanistic data and a few molecular related to clinical parameters data with some not fully addressed discrepancies.
Reviewer 3 Report
The manuscript entitled “Epigenetic mechanisms are involved in the oncogenic properties of ZNF518B in colorectal cancer” by Francisco Gimeno-Valiente et al., is primarily focused on determining the role of ZNF518B gene in colorectal cell dissemination and metastasis by epigenetic regulation of histone methyltransferases G9A and EZH2. Although the aim of this study is quite interesting, most of the experiments are exploratory in nature with inconclusive results. Since the expression of isoform 1 and 2 is not individually related to clinicopathological conditions of CRC patients, it’s difficult to understand the significance of i2/i1 in relapse only as shown in figure 1. Results shown in figure 2 include a contradictory message in terms of colony formation. It’s difficult to understand the legend for figure 3, which is completely missing. It is not worth mentioning ChIP analysis in section 2.5 as it didn’t work at all. Overall, the manuscript is writing is quite complex and difficult to determine a signification due to the exploratory nature of the studies.